



# Oblique rifting: the rule, not the exception.

Sascha Brune[1,2], Simon E. Williams[3], & R. Dietmar Müller[3,4]

[1] GFZ German Research Centre for Geosciences, 14473 Potsdam, Germany
[2] Institute of Earth and Environmental Science, University of Potsdam, 14476 Potsdam-Golm, Germany
[3] EarthByte Group, School of Geosciences, University of Sydney, NSW 2006, Australia
[4] Sydney Informatics Hub, University of Sydney, Sydney, NSW, Australia.

*Correspondence to*: Sascha Brune (brune@gfz-potsdam.de)

**Abstract.** Movements of tectonic plates often induce oblique deformation at divergent plate boundaries. This is in striking
contrast with traditional conceptual models of rifting and rifted margin formation, which often assume 2D deformation
where the rift velocity is oriented perpendicular to the plate boundary. Here we quantify the validity of this assumption by
analysing the kinematics of major continent-scale rift systems in a global plate tectonic reconstruction from the onset of
Pangea breakup until present-day. We evaluate rift obliquity by joint examination of relative extension velocity and local rift
trend using the script-based plate reconstruction software pyGPlates. Our results show that the global mean rift obliquity
amounts to 34° with a standard deviation of 24°, using the convention that the angle of obliquity is spanned by extension
direction and rift trend normal. We find that more than ~70% of all rift segments exceeded an obliquity of 20° demonstrating
that oblique rifting should be considered the rule, not the exception. In many cases, rift obliquity and extension velocity
increase during rift evolution (e.g. Australia-Antarctica, Gulf of California, South Atlantic, India-Antarctica), which suggests
an underlying geodynamic correlation via obliquity-dependent rift strength. Oblique rifting produces 3D stress and strain
fields that cannot be accounted for in simplified 2D plane strain analysis. We therefore highlight the importance of 3D
approaches in modelling, surveying, and interpretation of most rift segments on Earth where oblique rifting is the dominant
mode of deformation.

## 1 Introduction

The relative motion of Earth's tectonic plates often causes oblique deformation at divergent plate boundaries. This is
primarily due to the fact that irregularly shaped plate boundaries generally do not align with small-circles of relative plate
movement and that changes in plate motion additionally lead to time-dependent plate boundary obliquity (Philippon and
Corti, 2016; Díaz-Azpiroz et al., 2016). Traditionally, rift evolution and passive margin formation has been investigated
using 2D conceptual and numerical models assuming an alignment of relative plate motion and plate boundary normal.
These studies yielded major insights into first-order subsidence patterns (McKenzie, 1978; White, 1993), described key
phases controlling the architecture of rifted margins (Lavier and Manatschal, 2006; Huismans and Beaumont, 2011; Brune et




al., 2016) and provided insight into the fault evolution during rifting (Ranero and Pérez-Gussinyé, 2010; Brune et al., 2014. The applicability of these concepts and models is often rooted in the assumption that rifts can be understood via plane strain cross-sections orthogonal to the rift trend and that the direction of extension aligns with the orientation of these cross-sections. Many rifts and passive margins however involve segments where the extension direction is not perpendicular to the

rift strike such that oblique, non-plane strain configurations occur (Sanderson and Marchini, 1984; Dewey et al., 1998).

Oblique rift segments differ from classical orthogonal examples in several major aspects:

1) In contrast to orthogonal rifts, the initial phase of oblique rifting is characterized by segmented en-échelon border faults that strike transversely to the rift trend. The orientation of these faults is controlled by the interplay of

inherited heterogeneities with far-field stresses, whereas diverse modelling studies independently suggest a fault orientation that lies in the middle between the rift trend and the extension-orthogonal direction (Withjack and Jamison, 1986; Clifton et al., 2000; Agostini et al., 2009; Brune and Autin, 2013). These large en-échelon faults generate pronounced relay structures (Fossen and Rotevatn, 2016) which act as a major control on fluvial sediment transport (Gawthorpe and Leeder, 2000).

2) Seismic cross sections at rifted margins are often taken perpendicular to the rift trend, which in most cases is also perpendicular to the coastline. Considering that faults in oblique rifts do not strike parallel to the rift trend, seismic profiles will observe a projection of the fault surface that features a smaller dip angle than the actual 3D fault, an error that might cascade further into seismic restorations.

3) Besides these structural-observational implications, oblique rifting appears to be a major factor in governing the

geodynamic evolution of extensional systems. This has been shown via several lines of evidence: (i) Oblique rifting has been inferred to enhance strain localisation by enabling the formation of pull-apart basins and large-offset strike-slip faults, for instance during the Gulf of California opening (Bennett and Oskin, 2014; van Wijk et al., 2017). (ii) Analytical and numerical modelling suggests that the force required to maintain a given rift velocity is anti-correlated with the rift's obliquity. The reason for this behaviour is that plastic yielding takes place at smaller

tectonic force when the extension is oblique to the rift trend (Brune et al., 2012). (iii) At the same extension velocity, oblique rifts deform with a certain rift parallel shear rate, which is balanced by a lower rift-perpendicular extension velocity. This means that oblique segments of a particular rift accommodate lithospheric and crustal thinning at a lower rate than orthogonal segments of the same rift, a difference that effects the thermal configuration and therefore the structural and magmatic evolution of each segment (Montési and Behn, 2007).

4) Oblique rifting holds geodynamic implications on the global scale because of its relation to toroidal plate motion, i.e. vertical axis rotation components of plate movements and associated strike-slip deformation. Toroidal motion is enigmatic from the perspective of plate driving forces, because its purely horizontal motion cannot be directly caused by buoyancy forces in Earth's interior (Lithgow-Bertelloni et al., 1993; Bercovici, 2003). Oblique rifts



feature strike-slip velocity components and therefore contribute to toroidal motion, while orthogonal rifts are an expression of purely buoyancy-driven (poloidal) flow.

The impact of rift obliquity on the structural architecture of continental extensional systems varies between natural cases.
This is mainly due to rift variability in general, which arises from tectonic inheritance (Manatschal et al., 2015; Morley, 2016; Hodge et al., 2018; Phillips et al., 2018), or from along-strike changes in rheology, crustal configuration, temperature and rift velocity (e.g. Sippel et al., 2017; Molnar et al., 2017; Brune, et al., 2017b; Mondy et al., 2017).

Oblique rifting in presently active rifts can be easily deduced by combining the local rift trend and GNSS-based surface
velocities (Díaz-Azpiroz et al., 2016). Prominent examples are the Main Ethiopian rift (Corti, 2008), the Levant rift system including the Dead Sea rift (Mart et al., 2005), the Gulf of California rift (Atwater and Stock, 1998; Fletcher et al., 2007) and the Upper Rhine Graben (Bertrand et al., 2005). Structure and kinematics of past rift systems has been studied by surveying obliquely rifted margins (Fournier et al., 2004; Lizarralde et al., 2007; Klimke and Franke, 2016) and transform continental margins (Basile, 2015; Mericer de Lépinay et al., 2016). However, quantifying rift obliquity of a specific rift system through
time is more difficult since the syn-rift velocity evolution needs to be reconstructed from available geophysical and geological data sets. Therefore, a global statistical analysis of rift obliquity through geological time has been missing so far.

Here we strive to fill this gap by deducing the first-order obliquity history of Earth's major rifts from the onset of Pangea fragmentation to present-day. We first describe the applied methods and data sets, then we focus on major individual rift
systems that lead to the formation of the Atlantic and Indian Ocean basins before we assess rift obliquity evolution and average obliquity on a global scale.

## 2 Methods and data

### 2.1. Rift kinematics

We quantify extension velocity using the global kinematic plate reconstruction of Müller et al. (2016). This plate model
integrates the latest syn-rift reconstructions for the South Atlantic (Heine et al., 2013), North Atlantic (Barnett-Moore et al., 2016; Hosseinpour et al., 2013), Australia-Antarctica separation and India-Antarctica breakup (Williams et al., 2011; Whittaker et al., 2013; Gibbons et al., 2015), as well as Gulf of California opening (McQuarrie and Wernicke, 2005) among others (Figure 1).





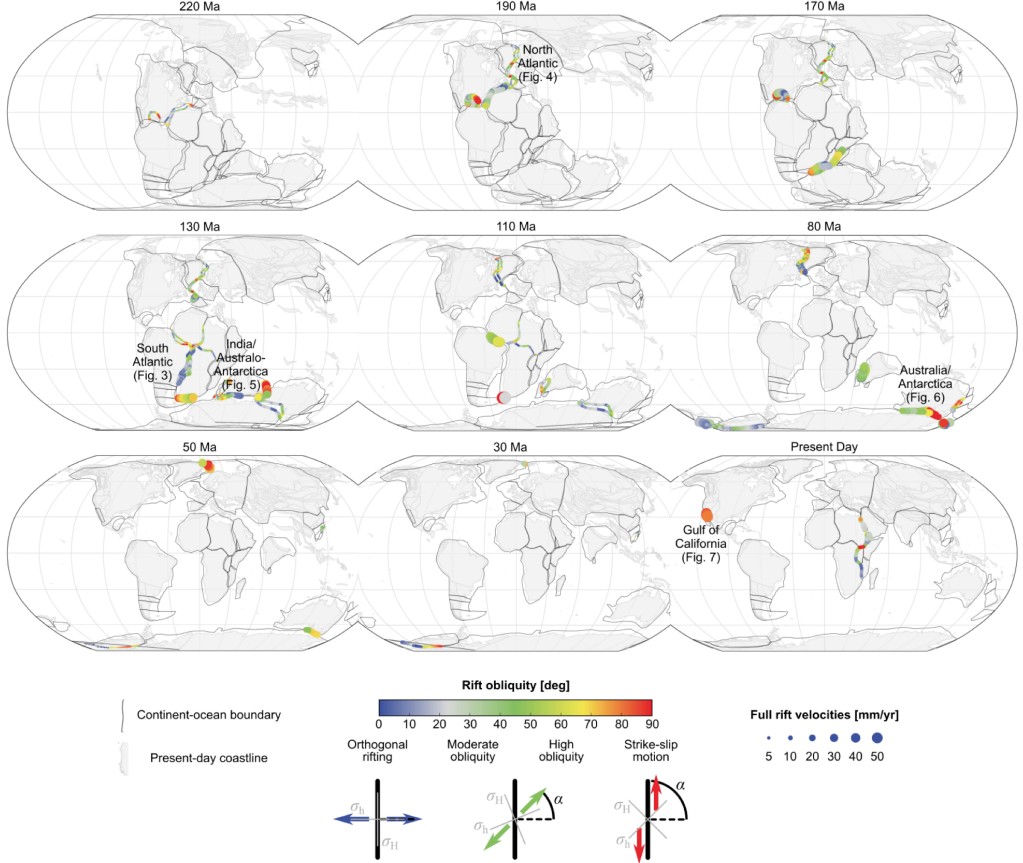

**Figure 1. Global overview of rift obliquity and velocity.** Rift obliquity is measured as angle $\alpha$ spanned by the relative plate velocity vector and the rift trend normal. The impact of rift obliquity on the rotation of largest and smallest horizontal stress components ($\sigma_H$ and $\sigma_h$, respectively) is depicted at the bottom of the figure. Rotations from Müller et al. (2016). Continent-ocean boundaries from Brune et al. (2016).

Restoration of the relative position of continents prior to rifting in aforementioned regional studies is largely based on deriving the amount of syn-rift extension from present-day crustal thickness (e.g. Williams et al., 2011; Kneller et al., 2012). The kinematic evolution before breakup, i.e. prior to the occurrence of oceanic fracture zones and oceanic magnetic anomalies, has to be inferred via careful joint interpretation of several geological indicators. Rift initiation for instance can be constrained through the ages of syn-rift sediments and rift-related volcanism, which give a minimum age for the beginning of rifting. Later syn-rift kinematics can be inferred from seismic tectono-stratigraphy and dating of rocks that have



been drilled or dredged within the continent–ocean transition, while additional information can be derived from kinematic indicators at neighbouring plate boundaries.

## 2.2. Rift trend

We define the rift trend as the general direction of a rift segment. Considering a typical rift width of ~100 km, the most

5   suitable wavelength to analyse rift trend variations is several hundred kilometres. Here we associate rift trend variations with the present-day boundary between continental and oceanic crust, often referred to as continent-ocean boundaries (COBs). COBs better reflect the actual rift trend than for instance present-day coastlines, since the latter are primarily affected by eustatic sea-level variations and local vertical motions (Figure 2a).

10   Considering COBs as sharp boundaries does not reflect the crustal complexity in these areas, which mirrors the convoluted interplay of tectonic, magmatic, and sedimentary processes. We therefore define two COB end-member sets that reflect the earliest and latest possible breakup based on available seismic refraction data (Brune et al., 2016). These COBs have simplified geometries and are designed to capture the regional geometry of boundaries between continental and oceanic crust

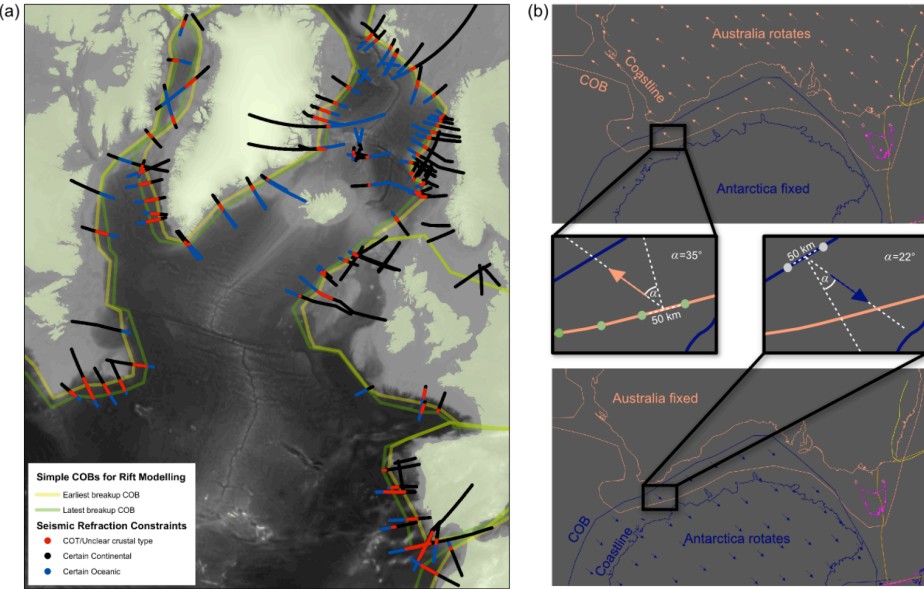

**Figure 2. Methods illustration.** **(a)** End-member sets of continent-ocean boundaries are based on seismic refraction data, here exemplified for the North Atlantic. The latest breakup COB constitutes our reference model while the earliest breakup COB is used for robustness tests. **(b)** Kinematic analysis approach exemplified for Australia-Antarctica rifting.



without a detailed assimilation of local data reflecting fine-scale deviations from these regional trends. Figure 2a depicts both the early and the late breakup COB data sets exemplified for the North Atlantic illustrating that the overall direction of the rift trend is not affected by the precise location of the COB.

The employed reconstruction is using rigid plate polygons and does not directly capture plate boundary deformation. Thus, in a pre-rift reconstruction, present-day COBs from conjugate passive margins will show significant overlap (Figure 1), where the amount of overlap is a proxy for the subsequent extension before breakup. As the plates move apart the overlap decreases, and the moment within the reconstructions where there is no longer overlap in each segment defines the transition from rifting to seafloor spreading. Within this methodology, the two central implications of how we interpret COB
geometries for our purposes is that they control the time of breakup along the margins, and that they define the orientation of each rift segment.

### 2.3 Rift obliquity

Using the COB orientation as a proxy for the rift trend, and accounting for the local direction of relative plate motions, we calculate rift obliquity for all points within an active rift at any time during post-Pangea extension. This workflow is
illustrated in Figure 2b, where we first keep Antarctica fixed in order to evaluate the rift obliquity at an Australian COB location and secondly we fix Australia and estimate rift obliquity for a point at the Antarctic COB. Note that the conjugate values for local rift obliquity are very similar but not the same. We therefore average obliquity values from conjugate margins during our statistical analysis. This analysis is repeated in 1 million year intervals until the conjugate COBs do not overlap anymore and the tectonic formation of the rifted margin ends.

We apply our workflow in an automated way using the python library pyGPlates that allows script-based access to GPlates functionality. GPlates is a free plate reconstruction software (www.gplates.org) that allows reconstructing and analysing plate motions through geological time (Müller et al., 2018). In the following, we use a spacing of 50 km between individual sample points where we extract relative plate velocity and obliquity. That spacing is dense enough to capture the relevant
changes in rift trend. We also tested smaller point distances, which did not affect our conclusions.

The limitations of this analysis workflow coincide with the limits of plate tectonic reconstructions in general. Many rifts and especially failed rifts are not included in plate tectonic reconstructions yet, which somewhat biases our study towards rifted margins. Future improvements in plate tectonic reconstructions and in defining COBs will enhance our results, however, by
testing several end-member scenarios in section 4 we can already anticipate that our main conclusions will still hold even though the detailed values might change.





In this study we follow the convention that defines the angle of obliquity as the angle between extension direction and local rift trend normal. This means that 0° represents orthogonal rifting while 90° stands for strike-slip motion. Note that this definition follows many previous studies (e.g. Fournier and Petit, 2007; Philippon et al., 2015; Brune, 2016; Zwaan and Schreurs, 2017; Ammann et al., 2017), but is opposite to the convention used in almost as many articles (e.g. Tron and Brun,
1991; Teyssier et al., 1995; Clifton and Schlische, 2001; Deng et al., 2018).

There is clearly a gradual transition from orthogonal rifting to oblique extension, especially since individual fault evolution is subject to natural variability. In this study, however, where we investigate the frequency of oblique rifting, it appears to be useful to draw a line between orthogonal and oblique extension. In simplified model settings, previous studies suggested that
qualitative differences in the rifting style emerge when rift obliquity exceeds 15 to 20° (Clifton et al., 2000; Agostini et al., 2009; Brune, 2014; Zwaan et al., 2016). Keeping in mind that the specific value is somewhat arbitrary, we will use 20° as the critical obliquity separating orthogonal from oblique rifts.

### 3 Regional analysis of individual rift systems

In this section our plate tectonic analysis is employed focussing on individual post-Pangea rift systems (Fig. 1). In doing so,
we relate the structural history of each rift with its obliquity and velocity evolution.

### 3.1. South Atlantic Rift

The orientation of the different South Atlantic Rift segments has been affected by reactivation of mobile belts, which formed during the pan-African orogeny in the Neoproterozoic and early Paleozoic (Kröner and Stern, 2004). This reactivation of inherited weaknesses is an ubiquitous process of the Wilson cycle that has also been evoked to explain the formation of the
present-day East African Rift System (Daly et al., 1989; Hetzel and Strecker, 1994). It has been suggested that 65% of the South Atlantic Rift developed with near-parallel orientation to the pan-African fabric (de Wit et al., 2008) evidencing the strong control of tectonic inheritance on the South Atlantic rift obliquity.

We find that low obliquity predominates in the central and southern segment of the South Atlantic rift (Figure 3). High rift
obliquities are encountered in the Equatorial Atlantic and in the southernmost shear zone that ultimately develops into the Falkland-Agulhas fracture zone. Our analysis shows that this system initially features a wide range of rift obliquities (Figure 3b) and trends to higher obliquity after 120 million years ago (Ma) when only the northern and southernmost segments are active. The frequency diagram of rift obliquity displays a bimodal distribution with the 0-25° range representing the southern South Atlantic while the 45-65° range is dominated by Equatorial South Atlantic rift geometry (Figure 3c). The overall mean
obliquity at 38° lies in–between these two peaks while 68% of the rift formed at moderate to high rift obliquity larger than 20°.





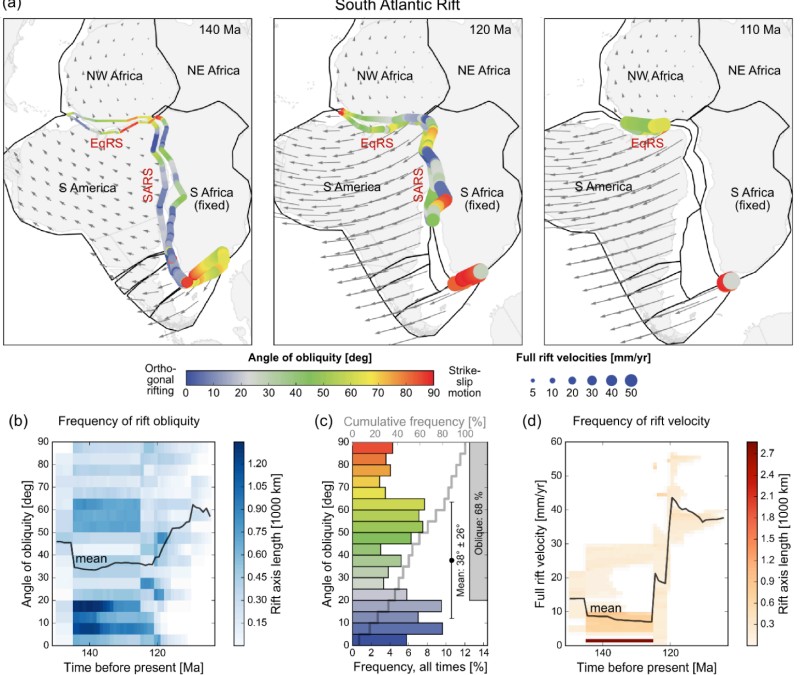

**Figure 3. South Atlantic rift.** Low rift obliquity prevails in the central and southern segment of the South Atlantic rift whereas high obliquities and strike-slip motion dominates the northern and southernmost segments. Rift velocity and obliquity increase jointly starting at 120 Ma. EqRS – Equatorial Atlantic Rift System; SARS – South Atlantic Rift System.

The rotational rifting of the South Atlantic with an initial Euler pole close to West Africa leads to higher rift velocities in the south than in the north. The mean rift velocity of the entire rift displays an initial, slow phase lasting more than 20 million

5   years, followed by rift acceleration during a few million years and finally a fast phase of rifting prior to the transition to sea-floor spreading (Heine et al., 2013; Brune et al., 2016). This two-phase evolution can also be seen in alternative reconstructions (Nürnberg and Müller, 1991; Torsvik et al., 2009; Moulin et al., 2010; Granot and Dyment, 2015) although it is partitioned between individual South American blocks for some studies (Brune et al., 2016). Interestingly, the evolution of rift obliquity and velocity appears to be correlated. On one side this shows that oblique segments need more time to reach

10   breakup since the effective rift-perpendicular velocity is smaller than in neighbouring purely orthogonal rift segments. To a certain degree, however, this might be due to the fact that oblique rifting requires less tectonic force, which leads to a higher rift velocity at constant extensional stress (Brune et al., 2012).



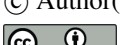

### 3.2. North Atlantic Rift

The formation of the North Atlantic involved a protracted rift history involving several major plates (Eurasia, Greenland, Iberia, North America). Initial inherited weaknesses from the Caledonian orogeny have been reactivated in episodic continental extension that is recorded in Carboniferous to Permian basins (Doré, 1991; Lundin and Doré, 1997). The
Mesozoic is marked by extensive crustal thinning that lead to formation and abandonment of major rift arms like the Rockall Trough, the Porcupine, Orphan, Møre, and Faroe-Shetland Basin (Skogseid et al., 2000; Faleide et al., 2008; Peron-Pinvidic et al., 2013). However, Mesozoic rifting also induced continental breakup in the Iberia-Newfoundland segment, the Bay of Biscay, and the North Atlantic rift south of Greenland (Féraud et al., 1996; Nirrengarten et al., 2018; Tugend et al., 2014). Opening of the Labrador Sea and rifting between Greenland and Europe competed for many tens of million years (Dickie et
al., 2011; Hosseinpour et al., 2013; Barnett-Moore et al., 2016) before the present-day North East Atlantic mid-ocean ridge formed in Eocene times (Gernigon et al., 2015; Gaina et al., 2017) possibly due to the arrival of the Iceland hotspot (Coffin and Eldholm, 1992; Storey et al., 2007). Final separation between Greenland and Europe took place along the sheared margin of the Fram Strait in Miocene times ~17-15 Ma (Jakobsson et al., 2007; Knies and Gaina, 2008).

Multiple plate motion changes reflect the complex tectonic history of the North Atlantic during the last 200 million years. These changes translate to time-dependent rift obliquity in each of the major rift branches (Figure 4). Between 200 and 120 Ma, the rift branches east and south of Greenland deform at 35° to 60° rift obliquity which leads to a pronounced peak within this obliquity range in Figure 4b and 4c. This changes in the Early Cretaceous with the more northward movement of Greenland and generates almost orthogonal rifting between Greenland and Europe until breakup. Due to the northward
propagation of sea-floor spreading, the Iberia-Newfoundland rift is excluded from the later plate motion changes and hence formed during low extension obliquity. The latest stage of continental rifting is marked by more than 30 million years of high-obliquity shear between northern Greenland and northwest Europe.

The absolute frequency of rift obliquity is marked by two peaks at 0° and 45° (Figure 4c). While this bears some similarity
with the South Atlantic (Figure 3c), the underlying reason for these two distinct peaks is not linked to the different orientation of two rift branches (like in the South Atlantic), but is a result of Greenland's plate motion change at around 120 Ma (Figure 4a,b). Except for Iberia-Newfoundland and the Labrador Sea, which evolved at low obliquity, most rift branches experienced moderate or high rift obliquity during the entire rift history. More than 70% of the rifts involved an obliquity of more than 20° and the overall mean obliquity of the North Atlantic rift amounts to 34° (Figure 4c).





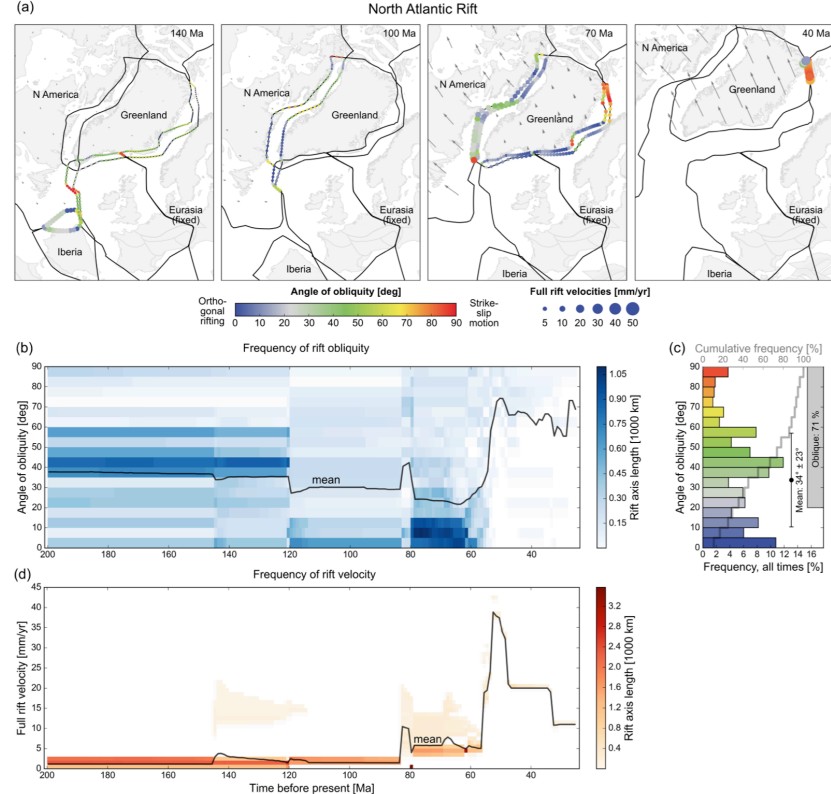

**Figure 4. North Atlantic rift.** Several changes in plate motion mirror the complex tectonic history of the North Atlantic during the last 200 million years. A distinct increase in syn-rift obliquity occurs at 50 Ma prior to final breakup between northern Greenland and northwest Europe.

### 3.3. Rifting between India and Australo-Antarctica

Mesozoic rifting within eastern Gondwana lead to continental fragmentation beginning with the separation of India (together with Sri Lanka and Madagascar) from Australia and Antarctic in the Early (Powell et al., 1988; Gibbons et al., 2013). The timing and kinematics of breakup and spreading between Australia and India is well constrained (Williams et al., 2013; Whittaker et al., 2016) and although the precise geometry of greater India is obscured by subsequent deformation during India-Eurasia collision, the divergence between India and Australia is thought to have involved a significant component of oblique motion along greater India's northern margin recorded by the Wallaby-Zenith Fracture Zone (Ali and Aitchison, 2014). Initial breakup between India and Antarctica is recorded in the Enderby Basin becoming progressively younger to the





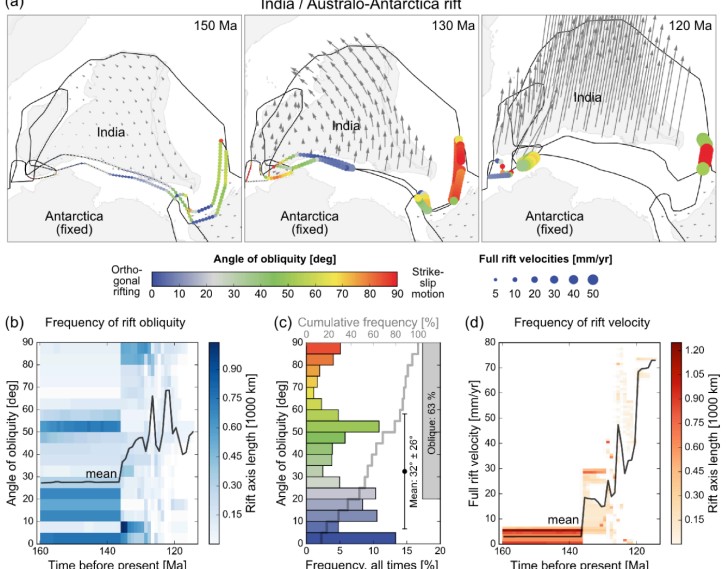

**Figure 5. India / Australo-Antarctica rift.** Rift obliquity is dominated by two major rift trends – the low obliquity India-Antarctica branch and the highly oblique India-Australia branch. For simplicity, we neglect low-velocity relative motion between Antarctica and Australia, which will be discussed in Figure 6.

west (Gibbons et al., 2013; Davis et al., 2016) and involved significant ridge jumps that isolated the Elan Bank microcontinent (Borissova et al., 2003).

Rotational rifting between East India and Australo-Antarctica with an Euler pole close to the southwestern tip of India
5  induces almost orthogonal rifting between India and Antarctica, and predominantly oblique rifting between India and West Australia (Figure 5). After 135 Ma, continued high obliquity shear along the northern Indian margin and Australia finally culminates in the formation of the Wallaby–Zenith Fracture Zone, the northern boundary of the Perth abyssal plain. At the same time breakup propagates from east to west along the East Indian margin, so that the mean rift obliquity is more and more dominated by high-angle shearing.

The rift obliquity distribution in Figure 5c reflects the existence of two major rift trends – the low obliquity India-Antarctica branch of 0° to 25° and the highly oblique India-Australia branch with more than 45°. The mean obliquity of the entire rift system amounts to 32° while more than 63% of the rift evolved at obliquity angles higher than 20°. Prior und during large parts of the breakup, the velocity and obliquity of the rift system are positively correlated.





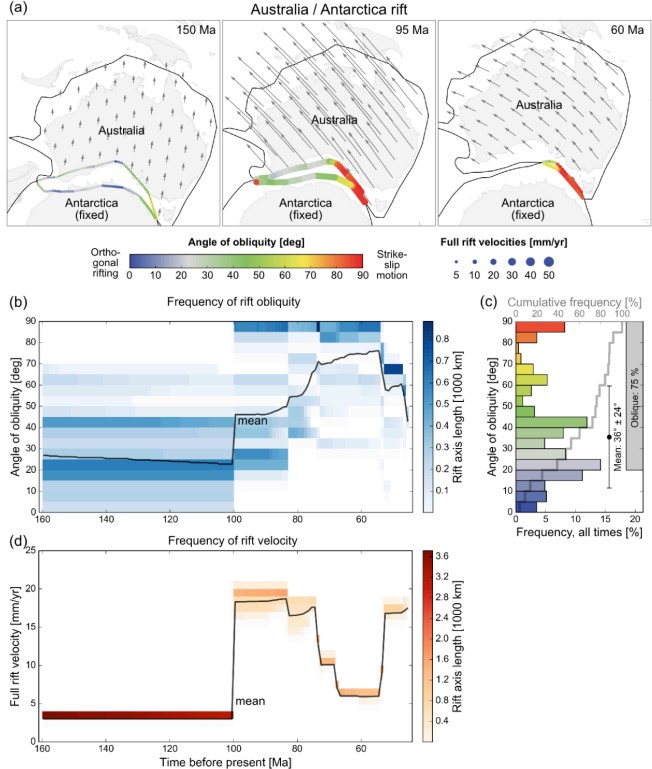

**Figure 6. Australia / Antarctica rift.** A distinct change in plate motion takes place at 100 Ma generating two discrete phases: (1) slow rifting at moderate obliquity, (2) fast rifting at high obliquity.

### 3.4. Australia / Antarctica

Phases of rifting between Australia and Antarctica are recorded beginning in the Late Jurassic (Powell et al., 1988; Ball et al., 2013) but continental breakup did not begin until the late Cretaceous and progressed diachronously from west to east.

5 Within this rift system, the signatures of oblique extension have been previously recognised along Australia's southern margin (Willcox and Stagg, 1990; Norvick and Smith, 2001). Lithospheric breakup was complex and protracted (Gillard et al., 2015, 2016), leaving unresolved questions surrounding the nature of the crust in the continent-ocean transition and the oldest interpreted seafloor spreading magnetic lineations. Reconstructions of the rift kinematics must therefore incorporate other geological constraints from along the Australia-Antarctica plate boundary system (Whittaker et al., 2013). This





reconstruction, in common with earlier studies (Powell et al., 1988; Royer and Sandwell, 1989), comprises an oblique component of divergence during the Late Cretaceous prior to a change in plate motion direction in the early Cenozoic.

The first-order history of rift obliquity can be understood by considering two distinct conditions. (i) Due to the concave
shape of the southern Australian margin and their Antarctic conjugates, there are two major rift trends along the Australia-Antarctica rift (Figure 6). (ii) According to the plate tectonic reconstruction (Williams et al., 2011), there has been a significant change in relative plate motion at around 100 Ma from a northward to a northwestward directed plate velocity. The two existing rift trends explain the dichotomy in rift obliquity of 10°-20° and 35°-45° from the onset of rifting until 100 Ma (Figure 6b). The plate motion change at 100 Ma however, shifts the rift obliquity in both branches to higher angles of
20°-45° and 80°-90°, respectively.

The velocity history displays a prominent increase at 100 Ma that corresponds to the increased rift obliquity via a plate motion change. An increase in the rate of plate divergence in the Late Cretaceous is corroborated by structural restoration studies based on seismic profiles (Espurt et al., 2012) and the rate of divergence is similar to that interpreted from initial
seafloor spreading anomalies (Tikku and Cande, 1999; Whittaker et al., 2013). Interestingly, after more than 20 million years of fast divergence, the relative plate velocity inferred from magnetic anomalies decreases again. If correct, this decrease cannot be related to Australia-Antarctica plate boundary dynamics, since at this time the rift system consists only of the last remaining continental bridge between Tasmania and Antarctica while the majority of the plate boundary already transitioned to sea floor spreading.

A further noteworthy aspect of Australia-Antarctic divergence is the failure of rifting between Tasmania and the southeast Australian mainland. Extension in the Bass and Gippsland basins occurred predominantly in the Early Cretaceous (e.g. Power et al., 2001). In the Late Cretaceous, rifting between Australia and Antarctica localised between western Tasmania and Cape Adare, where breakup eventually occurred. The higher obliquity of the successful plate boundary compared to the
failed rift arm in Bass Strait may explain why this rift was favoured, similar to the successful opening of the Equatorial Atlantic in favour of a Saharan Ocean during South Atlantic formation (Heine and Brune, 2014).

### 3.5. Gulf of California

The Gulf of California constitutes the youngest rift system in our analysis, which is why its temporal evolution is known in much greater detail than the previous case examples. The first phase of the Gulf of California rift is closely linked to the
Greater Basin and Range extensional zone. Tectono-stratigraphy and dated rift-related magmatic rocks show that the onset of continental extension must have occurred before the mid-Miocene (Ferrari et al., 2013; Duque-Trujillo et al., 2015). This first phase of slow rifting is marked by a wide rift style characteristic of the present-day Basin and Range region. An increase in both rift velocity and rift obliquity has been suggested as the underlying reason for basin-ward localisation and finally the




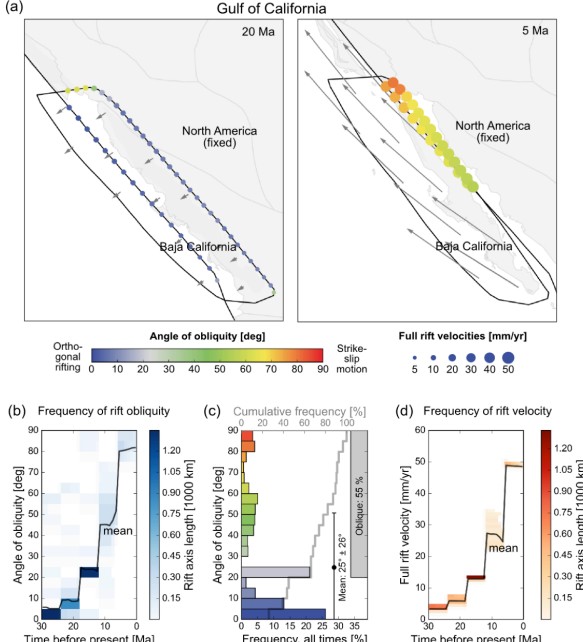

**Figure 7. Gulf of California rift.** The initial rift phase is characterised by slow, predominantly orthogonal rifting and associated with a wide rift. An increase in rift obliquity enhances localisation between 20 and 10 Ma inducing continental breakup. The obliquity and velocity of this rift increases almost proportionally.

transition to sea floor spreading in the southern Gulf of California (Bennett and Oskin, 2014; Darin et al., 2016; van Wijk et al., 2017). This change in plate motion at ~12 Ma has been explained by the final cessation of subduction in this region so that most of the relative plate motion between Pacific and North America had to be taken up by transform motion between Baja California and the North American mainland (Atwater and Stock, 1998; Oskin and Stock, 2003).

The transition from orthogonal to highly oblique rifting is captured by the tectonic reconstruction (McQuarrie and Wernicke, 2005) our plate model is built on (Figure 7a). According to that reconstruction, the transition occurs during a time frame of less than 10 million years between 20 and 10 Ma (Umhoefer, 2011). That transition is mirrored in our analysis by a gradual increase in rift obliquity from less than 10° prior to 18 Ma to 20-25° between 18 and 12 Ma up to 40-90° from 12 Ma until

10  present-day (Figure 7b).

A striking feature of the Gulf of California rift is that throughout the existence of this plate boundary the velocity evolved almost proportional to the rift obliquity (Figure 7b,d) hinting at a causal relationship between these two variables. Two





processes have been proposed to explain this proportionality: (i) Numerical and analytical modelling suggests that oblique extension requires less tectonic force than orthogonal rifting (Brune et al., 2012). The reason is that when extension is oblique to the rift trend, the plastic yield strength is reached at up to 50% less tectonic force. (ii) Localization within the lithosphere is enhanced by obliquity-related formation of pull-apart basins and associated energy-efficient strike-slip faults

(Umhoefer, 2011; Bennett and Oskin, 2014; van Wijk et al., 2017). Both processes apply to the Gulf of California and may in fact be simply different expressions of the same underlying cause, namely that oblique rifting is mechanically preferred.

## 4 Global analysis

In this section we evaluate global rift obliquity since the onset of Pangea fragmentation in terms of temporal and spatial variability. Analysing all rift systems of our global plate tectonic model during the last 230 million years, we here test the

robustness of our study by additionally considering the impact on passive margin area and by employing an alternative set of continent-ocean boundaries.

The extent of major rift systems varied through time (Figure 8a), with a pronounced peak between 160 and 110 million years ago when many rifts of the Atlantic and Indian Ocean where simultaneously active. Figure 8b illustrates that almost all

angles of obliquity are represented at any given time. Interestingly, obliquities in the range between 70° and 85° seem to be underrepresented while almost pure strike-slip systems are an ubiquitous feature. This finding might be explained by the fact that the transition from normal faulting to strike-slip faulting in ideal materials occurs at around 70° (Withjack and Jamison, 1986). We therefore speculate that once major continental strike-slip faults form, the plate boundary adjusts to a velocity-parallel configuration entailing the formation of a transform margin (Ammann et al., 2017), which also explains the

relatively high peak at 90° obliquity (Figure 8b,c).

In our reference model, we compute a mean global obliquity of 34° with a standard deviation of 24° since the inception of Pangea breakup. We also find that 69% of all rifts deform in oblique rift mode, i.e. with obliquities exceeding 20° (Figure 8c), illustrating that oblique rifting appears to be the rule on Earth rather than the exception.

So far, we analysed the frequency of obliquity with respect to rift length, where transform margins and rifted margins are given the same relative importance. Now we focus briefly on the surface area that is generated during passive margin formation, which is also a proxy for the size of sedimentary basins. We note that it is the rift-orthogonal velocity component, which leads to lithospheric stretching and generates passive margin area, while the rift parallel component merely induces an

along-strike offset. Hence we compute how many square kilometres of passive margin are generated per million year by multiplying the rift length of each small rift segment with its segment-orthogonal velocity component. The resulting distribution reflects the relative importance of rift obliquity weighted by passive margin area, which essentially leads to a





shift towards lower obliquity values. Nevertheless, we find an average obliquity of 29° and a total of 67% of passive margin area affected by obliquity angles larger than 20° (Figure 8e).

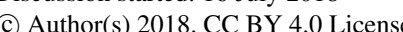

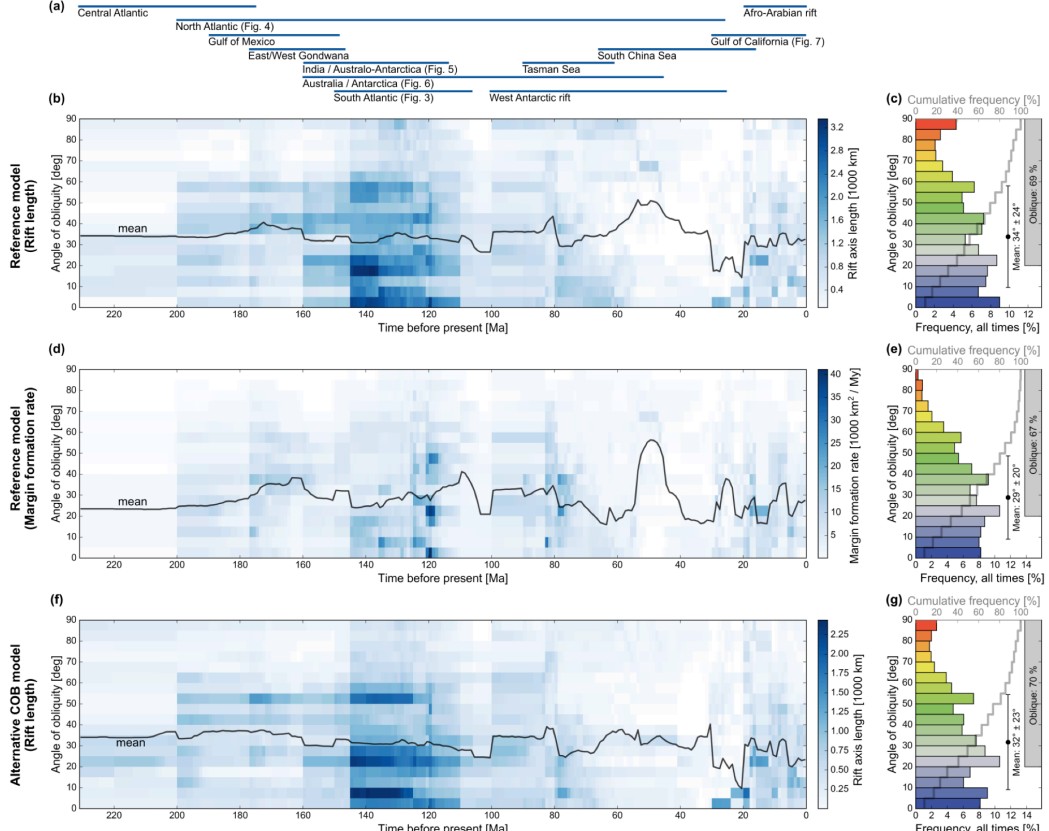

**Figure 8. Global analysis of rift obliquity. (a)** Variations of major rift system activity. Note that many rifts of the Atlantic and Indian Ocean where simultaneously active between 160 and 110 Ma. **(b,c)** Rift obliquity in terms of rift length for the reference model employing late-breakup COBs (see Sect. 2.2 and Fig. 2 for more details). **(d,e)** The reference model analysed in terms of margin formation rate (i.e. rift segment length multiplied with segment-orthogonal velocity component). **(f,g)** An alternative model employing early-breakup COBs. All three models result in global mean rift obliquities of ~30° and oblique rifting for more than ~70% illustrating the robustness of our results.



We test the impact of an alternative set of continent-ocean boundaries (Figure 2), which represents the continent-ward end member and hence stands for an earlier breakup time (Section 2.2). While the total rift length is reduced with respect to the reference model, we find that the first-order pattern of obliquity evolution is not affected. Also, the mean rift obliquity of 32° and the rift length percentage affected by oblique deformation (70%) is almost identical to the late-breakup end-member.

Finally, we map the time-averaged obliquity at each rift element (Figure 9). The advantage of this approach is that one can easily identify each point's rift obliquity that dominated the tectonic evolution, however, one has to keep in mind that changes in rift obliquity are not visualised. Figure 9 shows that only a few rifts exhibit a pure rift-orthogonal extension velocity such as the Labrador Sea, the east Indian margin and some locations in the North and South Atlantic. Instead, many

10  rifted margins feature moderate rift obliquity between 20° and 40° like the West Iberia margin, the Red Sea, as well as the central and southern segment of the South Atlantic. To a large extent, however, the dominant rift obliquity exceeds even 40°, for instance in the Gulf of Mexico, the Equatorial South Atlantic, the Gulf of Aden, the east African margins, the West Antarctic Rift, the Tasman Sea and also the sheared margins of the Fram Strait, Madagascar, and Western Australia.

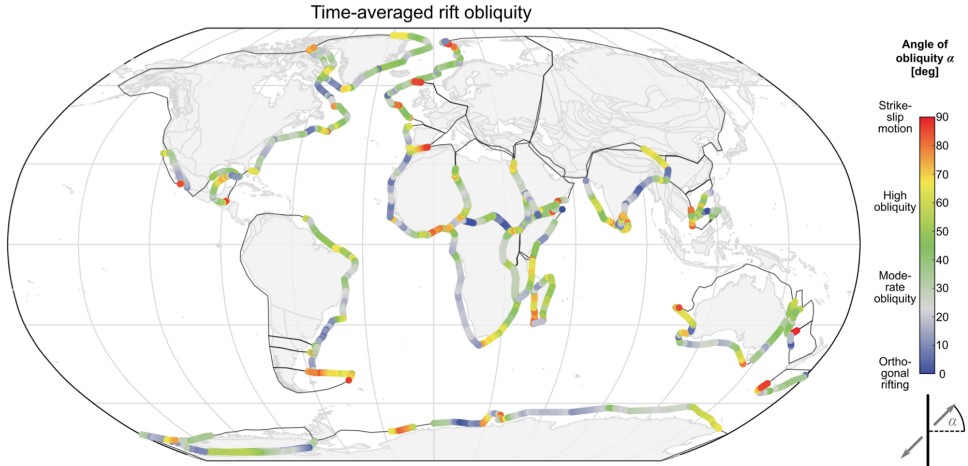

**Figure 9. Global map of mean rift obliquity.** For each rift point we display the time-averaged rift obliquity illustrating the prevalence of oblique rifting since Pangea fragmentation. Note that temporal changes of rift obliquity cannot be visualised in this plot.



## 5 Discussion

Previous studies quantified the present-day plate boundary obliquity in general terms and on long wavelength by considering extensional, compressional, and transform plate boundary types. Woodcock (1986) noted that 59% of all present-day plate boundaries feature obliquities larger than 22 degrees. A more detailed recent study found even higher obliquity by showing

that 65% of present-day plate boundaries exhibit >30° obliquity (Philippon and Corti, 2016). In the latter study, this result was further decomposed by plate boundary type illustrating that 73% of rifts and mid-ocean ridges extend at more than 30° obliquity. While these previous studies did not focus on rift obliquity, they nevertheless are consistent with our results by highlighting that oblique plate boundary deformation constitutes the rule and not the exception.

Jeanniot and Buiter (2018) quantified the margin width of transtensionally formed rifted margins and thereby estimated the rift obliquity of 26 major rift segments worldwide. They find a weak positive correlation between obliquity and width of a rift system; however, at highly oblique margins this relationship breaks down and these margins are not only significantly narrower than orthogonal margins, but they also exhibit large-offset transform faults. In contrast to our approach, Jeanniot and Buiter (2018) did not focus on the temporal evolution of rift obliquity. Nonetheless, for the time-averaged obliquity

(Figure 9) we find a very good correspondence with their results illustrating the robustness of both approaches.

In general terms, oblique deformation takes place whenever a plate boundary exhibits irregular trends (Dewey et al., 1998) and is therefore an obvious ingredient of plate boundary tectonics. However, we speculate that the unexpectedly high percentage of oblique rifts and rifted margins could also reflect the existence of an underlying geodynamic process that

influences plate velocities and thereby affects the resulting obliquity distribution. Previous analogue and numerical models suggest oblique rifting as a mechanically preferred type of continental extension (Chemenda et al., 2002; Brune et al., 2012). The reason for this behaviour has been deduced by means of analytical modelling suggesting that plastic yielding takes place at less tectonic force when the relative plate velocity is oblique to the rift trend (Brune et al., 2012) exerting additional control on rift strength that is otherwise governed by thermo-rheological properties, strain localisation and inherited

weaknesses (Burov, 2015; Buck, 2015; Brune, 2018). As a consequence, during rift competition, oblique rifting should prevail over orthogonal rifting if all other rift parameters are similar. This appears to be the case for the West African and Equatorial Atlantic rift systems, which were active at the same time until localisation along the more oblique Equatorial Atlantic rift induced the failure of the West African rift (Heine and Brune, 2014). The same process could explain the kinematic evolution of the Australia-Antarctica and Gulf of California rifts. In both cases a change in the direction of plate

divergence induced higher rift obliquity and simultaneously the rift velocity increased. We suggest that following the change in extension direction the higher localisation efficiency of oblique rifts sparked a significant loss in rift strength (Brune et al., 2016), which ultimately generated a speed-up of Baja California and Australia.





Oblique rifting is closely linked to toroidal plate motion, i.e. the spin of plates and associated strike-slip deformation. The concept of decomposing Earth's plate motions into toroidal (plate boundary parallel) and poloidal (plate boundary perpendicular) components is motivated by the insight that toroidal motion does not affect the buoyancy configuration of Earth's mantle. It is hence not directly driven by mantle convection and in a homogeneous plate-mantle system, energy

consumption due to toroidal motion should therefore be minimized (O'Connell et al., 1991). Large lateral rheological contrasts within and between Earth's plates have been invoked to explain some part of the toroidal motion component (Becker, 2006; Rolf et al., 2017). However, an additional part of the toroidal plate motion might be due to rift obliquity: since oblique rifting reduces rift strength, it favours the development of oblique plate boundaries, which enhances large-scale plate rotation and associated toroidal surface velocity components. Lithgow-Bertelloni and Richards (1993) showed that the

toroidal component of plate motions slowly declined since 120 Ma, despite large variations in the poloidal component of plate velocities. Notwithstanding significant uncertainty, the ratio of toroidal/poloidal velocities appears to be especially high between 120 and ~80 Ma. This period corresponds to a distinct decline in the lengths of rifts involved in Pangea breakup (Fig. 8, and also Brune et al., 2017c). Hence we speculate that the toroidal/poloidal ratio could be higher during continental breakup because of the rift characteristic to favour oblique motion, a process that has less impact on plate motions once the

continents become dispersed.

### 6 Conclusions

The 2D assumption that the extension direction is perpendicular to the rift trend is not justified in most cases. We find that the majority of rift systems leading to continental breakup during the last 230 million years involved moderate to high rift obliquity. Approximately 70% of all rift segments involved a distinct obliquity higher than 20° while the global average in

terms of rift obliquity is 34°. This high contribution of oblique deformation can be explained through the generally irregular shape of plate boundaries, possibly related to tectonic inheritance, and by the concept of obliquity-dependent plate boundary strength. Oblique deformation generates intrinsically 3D stress and strain fields that hamper simplified tectonic interpretation via 2D cross sections, models and seismic profiles. Our results indicate that oblique plate boundary deformation should be considered the rule and not the exception when investigating the dynamics of rifts and rifted margins.

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
