# Peer review of "Oblique rifting: the rule, not the exception."

_Solid Earth, 2018_

## Referee Comment (RC1) · Anonymous Referee #1 · 8 Aug 2018

In this paper the Authors investigate the obliquity of major continental rift systems from the onset of Pangea breakup to present by using global plate reconstructions. In particular they quantify rift obliquity by analysing the local extension direction and the assumed rift trend by using the sotware pyGPlates. The Authors find that the majority of rift systems are oblique by more than 20°, therefore suggesting that rift obliquity is the rule, not the exception. This has important implications for the interpretation of most rift systems on Earth, for which a complex 3D evolution must be considered.

Overall, I enjoyed reading the manuscript, which is well written, illustrated and clear; it offers very interesting insights into the process of continental rifting. I therefore support its publication. I only have some minor suggestions, which could improve this intersting work, and which are listed below.

Pag,2, Line 9. I suggest not to use the term 'transversely' here. It can be some-

[Figure]

how misleading - 'transversal' is normally used to indicate structures trending almost orthogonal to the rift trend. I suggest to replace it with 'obliquely' or similar

Pag,2, Line 12. Maybe a referece to the work by Corti 2008 could be appropriate here

Caption Figure 1. Instead of using the notation sigma H and h, I suggest to clearly indicate sigma Hmin and max

Pag,2, Line 7. Some references here?

Pag 3. In section 2 (or maybe in the Discussion) the Authors could discuss in some more detail the similarities or differences with the methods used to calculate obliquity in previous works. I refer in particular to the work by Jeanniot and Buiter (2018), where a similar analysis is presented.

Pag. 7, Line 4. Reference to the work by Withjack and Jamison 1986 needed (before Tron and Brun, 1991)

Pag. 8, Lines 8 and following. The relations between rift velocity and obliquity are not very clear to me, and could be maybe discussed in some more detail. In particular, I maybe misunderstood something but the first explanation for this behaviour seems not to be consistent with observations (i.e., the higher the obliquity, the lower the velocity), so it is really not clear to me. Also note that the relation between obliquity and velocity is also repeated in section 3.5 (Gulf of California), at the beginning of Pag 15. In order to avoid these repetitions, and not to mix observations with explanation of results, I suggest to think about moving the interpretation of the correlation between velocity and obliquity to the Discussion section.

Pag. 9, Line 6. . . ..Faroe-Shetland basins..

Figure 4. In panel b, I guess the abscissa represents the Time before present, but it is not very clear in its present form (it seems indeed that the X-axis indicates the Frequency of rift velocity). Please check and fix this

[Figure]

Pag. 10, Line 4. Early ??? (something missing here)

---

## Referee Comment (RC2) · Anonymous Referee #2 · 21 Sep 2018

This paper presents a very interesting global quantitative study of rift obliquity and concludes that much more rift segments are oblique than previously admitted. This study is well documented and well described, with focus given on few well-known oblique rift examples. The introduction and discussion are more focused on the geodynamic implications and the mechanical reason that could explain their observations. I have only a few minor comments, the first one concerns the choice of the author to consider the COB as the rift trend to define obliquity which should be discuss a little bit more in view of existing analogue and numerical model and the second concerns the mechanical discussion at the end of the paper, which is too light and could be improved a lot.

First point : At line 5 page 5 or line 13 page 26 The authors state that the COB is an indicator of rift trend and use it to define obliquity by comparing with the direction of plate. It is a choice, which is not better nor worse than another choice but it is arguable, and I would like to see a bit more argumentation in the text than it is a better choice

than the coast line. The COB is an indicative of the trend of the rift in the lithospheric mantle that is in the very late stage of rifting after the mantle necking. 3D models show that the COB might have a different obliquity then the initial trend and that with time obliquity of rift structure might increase as the lithosphere become weaker and weaker or decrease.

Second point: Based on a previous paper by the first author, this contribution explains the predominance of obliquity is that oblique rift requires less forces than orthogonal rifting. I think this analysis is biased and not complete because 1/ oblique rifts only emerges when transtension is imposed either by the BC's or the initial weak zones (thermal or mechanical) imposed on models. If their statement was indeed true, models of propagating rift segment would all produce oblique rift and none of them do (Van Vijk and blackmann 2005, Allken et al. 2012, Mondy et al 2017, Le Pourhiet et al. 2018) actually do unless a oblique weak zone is imposed (Molnar et al. 2017 2018, Balazs et al. 2018) or propagation stagnates (Van Vijk and blackmann 2005, Allken et al. 2012, Le Pourhiet et al. 2018)

2/ the authors completely dismiss mechanical or thermo-mechanical models that simulates oblique rifting using the standard set up of Le Calvez, and Vendeville, (2002), that is: two offset weak zones embedded in a rectangular box with imposed cylindrical boundary conditions (now, renamed as transtensional by Zwaan et al. 2016). In these set up, the linkage zone forming between the weak zone in Allken et al 2012, Liao and Gerya 2016, Le Pourhiet et al. 2017 , Amman et al. 2017, or Balazs et al. 2018, are also an oblique rift segment in your definition (COB vs boundary condition). Yet, if ones consider, the surface area and the number of faults as a proxi, the work required to rift this oblique segment is much larger than to rift the orthogonal segment. The main difference between these set-ups is that in one case (Brune et al 2012) the there is a pre-existing linear weak zone in the mantle lithosphere (i.e. the necking zone is pre-imposed) while in the other, it forms self consistently when crust and mantle deformation couple (i.e. in the late stage of rifting).

I think, that given the expertise of the first author, a more thorough discussion of the role of heritage, rift propagation or lack of it, and linkage as possible explanation of the omnipresence of oblique margin could really improve the paper a lot without lengthening it by much.

Ammann, N., Liao, J., Gerya, T. and Ball, P., 2017. Oblique continental rifting and long transform fault formation based on 3D thermomechanical numerical modeling. Tectonophysics.

Allken, V., Huismans, R.S. and Thieulot, C., 2012. Factors controlling the mode of rift interaction in brittle‐ductile coupled systems: A 3D numerical study. Geochemistry, Geophysics, Geosystems, 13(5).

Brune, S., Popov, A.A. and Sobolev, S.V., 2012. Modeling suggests that oblique extension facilitates rifting and continental break‐up. Journal of Geophysical Research: Solid Earth, 117(B8).

Balázs, A., Matenco, L., Vogt, K., Cloetingh, S. and Gerya, T., 2018. Extensional polarity change in continental rifts: inferences from 3D numerical modeling and observations. Journal of Geophysical Research: Solid Earth.

Le Calvez, J.H. and Vendeville, B.C., 2002. Experimental designs to model along-strike fault interaction. Journal of the Virtual Explorer, 7, pp.1-17.

Le Pourhiet, L., Chamot-Rooke, N., Delescluse, M., May, D.A., Watremez, L. and Pubellier, M., 2018. Continental break-up of the South China Sea stalled by far-field compression. Nature Geoscience, 11(8), p.605.

Le Pourhiet, L., May, D.A., Huille, L., Watremez, L. and Leroy, S., 2017. A genetic link between transform and hyper-extended margins. Earth and Planetary Science Letters, 465, pp.184-192.

Liao, J. and Gerya, T., 2015. From continental rifting to seafloor spreading: insight from 3D thermo-mechanical modeling. Gondwana Research, 28(4), pp.1329-1343.

Molnar, N.E., Cruden, A.R. and Betts, P.G., 2017. Interactions between propagating rotational rifts and linear rheological heterogeneities: Insights from three‐dimensional laboratory experiments. Tectonics, 36(3), pp.420-443.

Molnar, N.E., Cruden, A.R. and Betts, P.G., 2018. Unzipping continents and the birth of microcontinents. Geology, 46(5), pp.451-454.

Mondy, L.S., Rey, P.F., Duclaux, G. and Moresi, L., 2017. The role of asthenospheric flow during rift propagation and breakup. Geology, 46(2), pp.103-106.

Van Wijk, J.W. and Blackman, D.K., 2005. Dynamics of continental rift propagation: the end-member modes. Earth and Planetary Science Letters, 229(3-4), pp.247-258.

Zwaan, F., Schreurs, G., Naliboff, J. and Buiter, S.J., 2016. Insights into the effects of oblique extension on continental rift interaction from 3D analogue and numerical models. Tectonophysics, 693, pp.239-260.

---

## Author Comment (AC1) · 4 Oct 2018

Author's response to comments by Reviewer #1

Rev #1: In this paper the Authors investigate the obliquity of major continental rift systems from the onset of Pangea breakup to present by using global plate reconstructions. In particular they quantify rift obliquity by analysing the local extension direction and the assumed rift trend by using the software pyGPlates. The Authors find that the majority of rift systems are oblique by more than $20°$, therefore suggesting that rift obliquity is the rule, not the exception. This has important implications for the interpretation of most rift systems on Earth, for which a complex 3D evolution must be considered. Overall, I enjoyed reading the manuscript, which is well written, illustrated and clear; it offers very interesting insights into the process of continental rifting. I therefore support its publication. I only have some minor suggestions, which could

improve this interesting work, and which are listed below.

Authors: We thank Reviewer #1 for this motivating assessment. All suggestions have been addressed in the manuscript and our responses are listed below.

————————————————

Rev #1: Pag,2, Line 9. I suggest not to use the term 'transversely' here. It can be somehow misleading - 'transversal' is normally used to indicate structures trending almost orthogonal to the rift trend. I suggest to replace it with 'obliquely' or similar

Authors: Done.

————————————————

Rev #1: Pag,2, Line 12. Maybe a reference to the work by Corti 2008 could be appropriate here

Authors: Done.

————————————————

Rev #1: Caption Figure 1. Instead of using the notation sigma H and h, I suggest to clearly indicate sigma Hmin and max

Authors: Done.

————————————————

Rev #1: Pag,2, Line 7. Some references here?

Authors: References added (Chaboureau et al., 2013; Quirk et al., 2013).

————————————————

Rev #1: Pag 3. In section 2 (or maybe in the Discussion) the Authors could discuss in some more detail the similarities or differences with the methods used to calculate obliquity in previous works. I refer in particular to the work by Jeanniot and Buiter
(2018), where a similar analysis is presented.

Authors: We extended the comparison to Jeanniot and Buiter (2018) in the discussion section and highlighted similarities or differences of our approaches. See paragraph "Jeanniot and Buiter (2018) evaluated the margin width..." in section 5.

—————————————

Rev #1: Pag. 7, Line 4. Reference to the work by Withjack and Jamison 1986 needed (before Tron and Brun, 1991)

Authors: Reference added.

—————————————

Rev #1: Pag. 8, Lines 8 and following. The relations between rift velocity and obliquity are not very clear to me, and could be maybe discussed in some more detail. In particular, I maybe misunderstood something but the first explanation for this behaviour seems not to be consistent with observations (i.e., the higher the obliquity, the lower the velocity), so it is really not clear to me. Also note that the relation between obliquity and velocity is also repeated in section 3.5 (Gulf of California), at the beginning of Pag 15. In order to avoid these repetitions, and not to mix observations with explanation of results, I suggest to think about moving the interpretation of the correlation between velocity and obliquity to the Discussion section.

Authors: We formulated the relation between obliquity and velocity more clearly and moved it to the discussion section. See paragraph "In many cases, we find a correlation between the obliquity and the velocity of a rift..." in section 5.

—————————————

Rev #1: Pag. 9, Line 6. . . ..Faroe-Shetland basins..

Authors: Done.

————————————————

Figure 4. In panel b, I guess the abscissa represents the Time before present, but it is not very clear in its present form (it seems indeed that the X-axis indicates the Frequency of rift velocity). Please check and fix this

Authors: We agree and changed the axis labels of Figure 4. The same problem has been fixed in Figure 6.

————————————————

Rev #1: Pag. 10, Line 4. Early ??? (something missing here)

Authors: Fixed.

---

## Author Comment (AC2) · 4 Oct 2018

Author's response to comments by Reviewer #2

Rev #2: This paper presents a very interesting global quantitative study of rift obliquity and concludes that much more rift segments are oblique than previously admitted. This study is well documented and well described, with focus given on few well-known oblique rift examples. The introduction and discussion are more focused on the geodynamic implications and the mechanical reason that could explain their observations. I have only a few minor comments, the first one concerns the choice of the author to consider the COB as the rift trend to define obliquity which should be discuss a little bit more in view of existing analogue and numerical model and the second concerns the mechanical discussion at the end of the paper, which is too light and could be improved a lot.

[Figure]

Authors: We thank Reviewer #2 for these suggestions. In the revised version we discuss the continent-ocean boundary as a rift trend proxy and extend the paragraph concerning the underlying mechanical reasons of oblique rifting.

———————————————

Rev #2: First point : At line 5 page 5 or line 13 page 26 The authors state that the COB is an indicator of rift trend and use it to define obliquity by comparing with the direction of plate. It is a choice, which is not better nor worse than another choice but it is arguable, and I would like to see a bit more argumentation in the text than it is a better choice than the coast line. The COB is an indicative of the trend of the rift in the lithospheric mantle that is in the very late stage of rifting after the mantle necking. 3D models show that the COB might have a different obliquity then the initial trend and that with time obliquity of rift structure might increase as the lithosphere become weaker and weaker or decrease.

Authors: We added a discussion concerning the advantages and disadvantages of three potential rift trend proxies: COB, coastline and topographic highs. See paragraph "We define the rift trend as the general direction of a rift segment . . ." in section 2.2.

———————————————

Rev #2: Second point: Based on a previous paper by the first author, this contribution explains the predominance of obliquity is that oblique rift requires less forces than orthogonal rifting. I think this analysis is biased and not complete because 1/ oblique rifts only emerges when transtension is imposed either by the BC's or the initial weak zones (thermal or mechanical) imposed on models. If their statement was indeed true, models of propagating rift segment would all produce oblique rift and none of them do (Van Vijk and blackmann 2005, Allken et al. 2012, Mondy et al 2017, Le Pourhiet et al. 2018) actually do unless a oblique weak zone is imposed (Molnar et al. 2017 2018, Balazs et al. 2018) or propagation stagnates (Van Vijk and blackmann 2005, Allken et al. 2012, Le Pourhiet et al. 2018) 2/ the authors completely dismiss mechanical or

thermo-mechanical models that simulates oblique rifting using the standard set up of Le Calvez, and Vendeville, (2002), that is: two offset weak zones embedded in a rectangular box with imposed cylindrical boundary conditions (now, renamed as transtensional by Zwaan et al. 2016). In these set up, the linkage zone forming between the weak zone in Allken et al 2012, Liao and Gerya 2016, Le Pourhiet et al. 2017, Amman et al. 2017, or Balazs et al. 2018, are also an oblique rift segment in your definition (COB vs boundary condition). Yet, if ones consider, the surface area and the number of faults as a proxi, the work required to rift this oblique segment is much larger than to rift the orthogonal segment. The main difference between these set-ups is that in one case (Brune et al 2012) the there is a pre-existing linear weak zone in the mantle lithosphere (i.e. the necking zone is pre-imposed) while in the other, it forms self consistently when crust and mantle deformation couple (i.e. in the late stage of rifting). I think, that given the expertise of the first author, a more thorough discussion of the role of heritage, rift propagation or lack of it, and linkage as possible explanation of the omnipresence of oblique margin could really improve the paper a lot without lengthening it by much.

Authors: We agree with the reviewer that the initial and boundary conditions exert strong control on model evolution in these kinds of experiments. In the revised version we added a discussion on this point where we compare the different setups and potential causes of oblique rifting. But since the focus of our manuscript lies on the analysis of plate tectonic reconstructions and not on numerical modeling, we do not to discuss the results and implications of these models in more detail than necessary. See paragraph "The dynamics of oblique rifting have been previously investigated ..." in section 5.

---

## Editor Decision (ED1)

**Author's response**

This file contains our point-by-point responses to the reviews, and a marked-up manuscript version where all our changes are tracked in red.

**Author's response to comments by Reviewer #1**

Rev #1: In this paper the Authors investigate the obliquity of major continental rift systems from the onset of Pangea breakup to present by using global plate reconstructions. In particular they quantify rift obliquity by analysing the local extension direction and the assumed rift trend by using the software pyGPlates. The Authors find that the majority of rift systems are oblique by more than 20°, therefore suggesting that rift obliquity is the rule, not the exception. This has important implications for the interpretation of most rift systems on Earth, for which a complex 3D evolution must be considered. Overall, I enjoyed reading the manuscript, which is well written, illustrated and clear; it offers very interesting insights into the process of continental rifting. I therefore support its publication. I only have some minor suggestions, which could improve this interesting work, and which are listed below.

Authors: We thank Reviewer #1 for this motivating assessment. All suggestions have been addressed in the manuscript and our responses are listed below.
* * *
Rev #1: Pag,2, Line 9. I suggest not to use the term 'transversely' here. It can be somehow misleading - 'transversal' is normally used to indicate structures trending almost orthogonal to the rift trend. I suggest to replace it with 'obliquely' or similar

Authors: Done.
* * *
Rev #1: Pag,2, Line 12. Maybe a reference to the work by Corti 2008 could be appropriate here

Authors: Done.
* * *
Rev #1: Caption Figure 1. Instead of using the notation sigma H and h, I suggest to clearly indicate sigma Hmin and max

Authors: Done.
* * *
Rev #1: Pag,2, Line 7. Some references here?

Authors: References added (Chaboureau et al., 2013; Quirk et al., 2013).
* * *
Rev #1: Pag 3. In section 2 (or maybe in the Discussion) the Authors could discuss in some more detail the similarities or differences with the methods used to calculate obliquity in previous works. I refer in particular to the work by Jeanniot and Buiter (2018), where a similar analysis is presented.

Authors: We extended the comparison to Jeanniot and Buiter (2018) in the discussion section and highlighted similarities or differences of our approaches. See paragraph "Jeanniot and Buiter (2018) evaluated the margin width…" in section 5.

50 ------------------------------------

Rev #1: Pag. 7, Line 4. Reference to the work by Withjack and Jamison 1986 needed (before Tron and Brun, 1991)

55 Authors: Reference added.
* * *
Rev #1: Pag. 8, Lines 8 and following. The relations between rift velocity and obliquity are not very clear to me, and could be maybe discussed in some more detail. In particular, I maybe misunderstood 60 something but the first explanation for this behaviour seems not to be consistent with observations (i.e., the higher the obliquity, the lower the velocity), so it is really not clear to me. Also note that the relation between obliquity and velocity is also repeated in section 3.5 (Gulf of California), at the beginning of Pag 15. In order to avoid these repetitions, and not to mix observations with explanation of results, I suggest to think about moving the interpretation of the correlation between velocity and obliquity to the 65 Discussion section.

Authors: We formulated the relation between obliquity and velocity more clearly and moved it to the discussion section. See paragraph "In many cases, we find a correlation between the obliquity and the velocity of a rift…" in section 5.

70 ------------------------------------

Rev #1: Pag. 9, Line 6. . . ..Faroe-Shetland basins..

Authors: Done.

75 ------------------------------------

Figure 4. In panel b, I guess the abscissa represents the Time before present, but it is not very clear in its present form (it seems indeed that the X-axis indicates the Frequency of rift velocity). Please check and fix this

80
Authors: We agree and changed the axis labels of Figure 4. The same problem has been fixed in Figure 6.
* * *
85 Rev #1: Pag. 10, Line 4. Early ??? (something missing here)

Authors: Fixed.

90

**Author's response to comments by Reviewer #2**

95   Rev #2: This paper presents a very interesting global quantitative study of rift obliquity and concludes that much more rift segments are oblique than previously admitted. This study is well documented and well described, with focus given on few well-known oblique rift examples. The introduction and discussion are more focused on the geodynamic implications and the mechanical reason that could explain their observations. I have only a few minor comments, the first one concerns the choice of the

100   author to consider the COB as the rift trend to define obliquity which should be discuss a little bit more in view of existing analogue and numerical model and the second concerns the mechanical discussion at the end of the paper, which is too light and could be improved a lot.

Authors: We thank Reviewer #2 for these suggestions. In the revised version we discuss the continent-
105   ocean boundary as a rift trend proxy and extend the paragraph concerning the underlying mechanical reasons of oblique rifting.
* * *
Rev #2: First point : At line 5 page 5 or line 13 page 26 The authors state that the COB is an indicator
110   of rift trend and use it to define obliquity by comparing with the direction of plate. It is a choice, which is not better nor worse than another choice but it is arguable, and I would like to see a bit more argumentation in the text than it is a better choice than the coast line. The COB is an indicative of the trend of the rift in the lithospheric mantle that is in the very late stage of rifting after the mantle necking. 3D models show that the COB might have a different obliquity then the initial trend and that with time
115   obliquity of rift structure might increase as the lithosphere become weaker and weaker or decrease.

Authors: We added a discussion concerning the advantages and disadvantages of three potential rift trend proxies: COB, coastline and topographic highs. See paragraph "We define the rift trend as the general direction of a rift segment …" in section 2.2.
120   ------------------------------------

Rev #2: Second point: Based on a previous paper by the first author, this contribution explains the predominance of obliquity is that oblique rift requires less forces than orthogonal rifting. I think this analysis is biased and not complete because
125   1/ oblique rifts only emerges when transtension is imposed either by the BC's or the initial weak zones (thermal or mechanical) imposed on models. If their statement was indeed true, models of propagating rift segment would all produce oblique rift and none of them do (Van Vijk and blackmann 2005, Allken et al. 2012, Mondy et al 2017, Le Pourhiet et al. 2018) actually do unless a oblique weak zone is imposed (Molnar et al. 2017 2018, Balazs et al. 2018) or propagation stagnates (Van Vijk and
130   blackmann 2005, Allken et al. 2012, Le Pourhiet et al. 2018)
2/ the authors completely dismiss mechanical or thermo-mechanical models that simulates oblique rifting using the standard set up of Le Calvez, and Vendeville, (2002), that is: two offset weak zones embedded in a rectangular box with imposed cylindrical boundary conditions (now, renamed as transtensional by Zwaan et al. 2016). In these set up, the linkage zone forming between the weak zone
135   in Allken et al 2012, Liao and Gerya 2016, Le Pourhiet et al. 2017, Amman et al. 2017, or Balazs et al. 2018, are also an oblique rift segment in your definition (COB vs boundary condition). Yet, if ones consider, the surface area and the number of faults as a proxi, the work required to rift this oblique

segment is much larger than to rift the orthogonal segment. The main difference between these set-ups is that in one case (Brune et al 2012) the there is a pre-existing linear weak zone in the mantle lithosphere (i.e. the necking zone is pre-imposed) while in the other, it forms self consistently when crust and mantle deformation couple (i.e. in the late stage of rifting).

I think, that given the expertise of the first author, a more thorough discussion of the role of heritage, rift propagation or lack of it, and linkage as possible explanation of the omnipresence of oblique margin could really improve the paper a lot without lengthening it by much.

Authors: We agree with the reviewer that the initial and boundary conditions exert strong control on model evolution in these kinds of experiments. In the revised version we added a discussion on this point where we compare the different setups and potential causes of oblique rifting. But since the focus of our manuscript lies on the analysis of plate tectonic reconstructions and not on numerical modeling, we do not to discuss the results and implications of these models in more detail than necessary. See paragraph "
[revised manuscript text omitted]

---

## Author Response (AR2)

**Author's response to comments by editor**
**(red text in has been added to previous version)**

Dear Federico Rossetti,

thank you very much for your comments. In the revised version, we accounted for all of them (see below).

Best wishes,
Sascha Brune
* * *
Editor: The West Antarctic Rift System should be mentioned here. available structural and geophysical data document a major transition from othrogonal to oblique rifting during the Cenozoic

Authors: We introduced a reference: "Prominent examples are the Main Ethiopian rift (Corti, 2008), the Levant rift system including the Dead Sea rift (Mart et al., 2005), the Gulf of California rift (Atwater and Stock, 1998; Fletcher et al., 2007), the Upper Rhine Graben (Bertrand et al., 2005), and the Cenozoic West Antarctic Rift (Rossetti et al., 2003; Vignaroli et al., 2015; Granot and Dyment, 2018)."
* * *
Editor: where these symbols are indicated in this and following figures?

Authors: We extended the description of symbols and also of the diagrams in the caption of Fig. 3 and link to this description in all other Figures: "Figure 3. South Atlantic rift. (a) Tectonic reconstruction of continental polygons. The magnitude of the local relative plate velocity along the rift system is represented by circle size, rift obliquity by circle colour. Relative velocities of continents with respect to a fixed South Africa are depicted as grey vectors. All velocities are based on the global reconstruction of Müller et al. (2016) and analysed using pyGPlates. Low rift obliquity prevails in the central and southern segment of the South Atlantic rift whereas high obliquities and strike-slip motion dominates the northern and southernmost segments. EqRS – Equatorial Atlantic Rift System; SARS – South Atlantic Rift System. (b) Frequency of rift obliquity in terms of rift axis length. The colour displays the integrated length of all rift segments that deform at the same rift obliquity. (c) Cumulative distribution of rift obliquity throughout the entire rift event. Bar colour represents obliquity. (d) Time-dependent frequency of rift velocity in terms of rift axis length. Colour shows integrated length of all segments deforming at the same velocity. Note that rift velocity and obliquity increase jointly starting at 120 Ma."
* * *
Editor: How defined? In this and other figures: arrows? meaning? how these kinematic vectors are obtained?

Authors: We explained more about the velocities and the stress definitions in the caption of Fig. 1: "Figure 1. Global overview of rift obliquity and velocity. Rift obliquity is measured as angle $\alpha$ spanned by the relative plate velocity vector and the rift trend normal. Rift kinematics are deduced at sample points along the present-day continental boundaries with a spacing of 50 km. Rift velocity magnitude is represented by circle size (see scale in the lower right corner), rift obliquity by circle colour (see scale in the middle). The impact of rift obliquity on the rotation of largest and smallest horizontal stress components ($\sigma_{Hmax}$ and $\sigma_{hmin}$, respectively) is depicted at the bottom of the figure and is based on relations discussed in Brune (2014). Rotations from Müller et al. (2016). Continent-ocean boundaries from Brune et al. (2016). "
* * *
Editor: Australia should be indicated. these tectonic structures should be indicated in the figure 5.

Authors: Done.

[revised manuscript text omitted]